# Comparison of Mangrove Stand Development on Accretion and Erosion Sites in Ca Mau, Vietnam

**Linh Thuy My Nguyen [1,2], Hanh Thi Hoang [3], Han Van Ta [2] and Pil Sun Park [1,*]** 

1   Department of Forest Sciences, Seoul National University, Seoul 08826, Korea; linh.ntm@rcfee.org.vn
2   Research Institute for Forest Ecology and Environment, Vietnamese Academy of Forest Sciences, Hanoi 100000, Vietnam; han.stmtr@gmail.com
3   South Western Forest Research and Experimental Center, Vietnamese Academy of Forest Sciences, Ca Mau 98000, Vietnam; hoangthihanh0105@gmail.com
*   Correspondence: pspark@snu.ac.kr; Tel.: +82-2-880-4771

**Abstract:** Mangroves are adapted to coastal processes; however, mangrove species showed various responses to estuarian environments, leading to different structural characteristics at accretion and erosion areas. The species composition, structure and regeneration of mangrove forests were investigated to provide insight into mangrove forest development in response to shoreline accretion and erosion processes. The species composition and stand structure of mangrove forests were measured along the distance from the shoreline at accretion and erosion sites in Ca Mau, Vietnam. The hierarchical clustering of grouped stands based on species composition and tree size distribution was conducted. Grouped mangrove stands showed landward changes in species composition and stand structure from the shoreline ($p < 0.05$), reflecting the timescale of accretion or erosion at both accretion and erosion sites. Stand development patterns differed between accretion and erosion sites, and *Avicennia alba* and *Rhizophora apiculata* dominated seaward plots at accretion and erosion sites, respectively. Newer accreted sites were dominated by *A. alba*. Mangrove stands developed from dense *A. alba* dominant to *R. apiculata* dominant stands with increasing tree size at accretion sites. There were more species-colonized sites with a higher erosion rate or that were more recently eroded, implying that timescale of erosion and erosion rate affected species composition and regeneration on erosion sites. Accretion and erosion affected stand development of mangroves differently, implying that conservation and restoration strategies should be applied differently to accretion and erosion sites.

**Keywords:** species composition; stand structure; erosion rate; shoreline distance; hierarchical clustering

## 1. Introduction

Mangrove forests are a representative estuarine ecosystem and are distributed in the inter-tidal region of approximately 140,000 km$^2$ in the tropics and subtropics [1]. Mangrove forests provide numerous ecosystem services, including acting as coastal carbon sinks and habitats for diverse coastal fauna [2,3]. Mangrove ecosystems are responsive to estuarial processes such as drainage, channelization, siltation, hurricanes, and thermal loading, which are related to a mangrove's tolerance to flood, salinity, and temperature. Mangroves are naturally adapted to ordinary estuarine processes [4], whereas they are sensitive to environmental changes caused by disturbances [5,6]. Numerous studies suggest that the vulnerability of mangroves during growth and succession processes can be used as a principle indicator of coastal disturbances [7].

Accretion and erosion are typical geomorphological processes in coastal estuaries and are regulated by means of sediment exchanges. Suspended sediment with different particle sizes is

distributed in estuaries through transport or trapping, creating accretion or erosion and affecting vegetation in estuaries [8,9]. Mangrove forests develop by interacting with sediment dynamics and shifting mudbanks, controlling erosion or accretion along the shoreline [10,11]. Estuary channels and mangroves on tidal flats import and retain sediments. Sedimentation leads to a seaward expansion of new mangrove habitats [12]. Mangroves play a key role in formation, stabilization, and development of muddy banks [13,14], and control the spatial and vertical distribution of nutrients and sediment grain size in estuaries [15]. As such, the coastal marsh is maintained under the balance between accretion and erosion processes [16]. However, a rise in sea-level attributable to climate change [17], tsunami damage [18,19], dam construction [20], sand mining, and shrimp farming [21] have accelerated coastal erosion, thereby negatively affecting mangrove ecosystems [22]. Human activities such as the conversion of mangroves to aquaculture or agriculture were identified as the primary cause of global mangrove loss from the period 1996–2010, and the greatest proportion of mangrove loss was observed in Southeast Asia [23]. The loss of mangroves attributable to human activities has accelerated erosion rates since the early twentieth century to almost 50 m per year in the coastal area of southern Vietnam [24].

Mangrove distribution or growth was actively studied on accretion areas or shores [13]. After the removal of mangals, coastal erosion was reported in many areas. However, information regarding structural development of mangrove forests on naturally-retreating coasts is still limited [25]. Studies focusing on the dynamics of mangrove forest responses to coastal disturbances of accretion and erosion are mainly large-scale and analyse changes in mangrove areas using Landsat images [26–28]. Recent studies have used Normalized Difference Vegetation Index (NDVI) of mangrove canopies and geometric characteristics of mangroves to detect the distribution patterns of mangrove forests via accretion or erosion [26].

Responses of mangrove ecosystems to external forces result in zonal distribution, which could be classified as fringe forests, riverine forests, overwash forests, basin forests, and dwarf forests [4]. Mangroves are primarily regenerated from viviparous propagules, which are dispersed land—or sea—wards by tidal amplitude, waves, or even hurricanes [29]. The colonization of mangroves is controlled by site specific conditions of salinity, light, canopy gaps, forestry canopy, and sediment characteristics [30]. Each mangrove species has its own tolerance to environmental factors such as salinity, flooding, or shade, which results in species zonation in mangrove forests [31]. *Avicennia marina* and *A. officinalis* are distributed in a wide range of soil salinities, while *Ceriops decandra*, and *Excoecaria agallocha* are restricted to low salinity areas [32]. *Lumnitzera racemosa* colonize in soil with a wide range of pH and medium salinity [33]. *Sonneratia alba*, *R. apiculata*, *A. officinalis*, and *C. tagal* are more tolerant to higher tidal inundations than *Bruguiera cylindrica* and *Xylocarpus granatum* during the seedling stage [34]. *A. marina* is more tolerant to waterlogged soil than *B. gymnorrhiza*, *E. agallocha*, and *L. racemosa* and is considered to be one of the most foreshore species [35]. *Avicennia* and *Rhizophora* are dominant species in mangrove forests; however, they differ in colonization and stand structural development. The *Avicennia* species is commonly dominant in fringe forest areas, while the *Rhizophora* species dominates more landward areas in basin and transition forests [36,37].

This study aims to understand the structural development of mangrove forests in response to different levels of coastal erosion and accretion processes in Ca Mau, belonging to the Mekong River Delta in Vietnam. The study results are expected to provide a scientific foundation for proposing effective practices in mangrove management and to assist in mitigating the continual problems of erosion in the Mekong River Delta region of Vietnam.

## 2. Materials and Methods

### 2.1. Study Sites

The study was conducted at the Ca Mau semi-island, which is in the southernmost area of Vietnam. Ca Mau is a coastal area with three sides surrounded by the sea and a total of 254 km

coastline; the Gulf of Thailand is to the west and the East Sea is to the south and east. The inland area has low terrain, with an average 0.3–1.6 m elevation [38]. The area has a typical monsoon climate, with distinct wet and dry seasons. The wet season is from May to October, and the dry season is from November to April of the next year. Annual precipitation is 2366 mm, and 87.7% of the annual precipitation is estimated to occur during the wet season [39]. The annual mean relative humidity is 86%, and the annual mean temperature is 26.5 °C, with a maximum monthly mean temperature of 28 °C in April or May and minimum monthly mean temperature of 25.3 °C in January [38]. The soil in Ca Mau province has been classified into four groups: Fluvisols (231,983.1 ha), Gleysols (236,598.3 ha), Histosols (19,209.4 ha), and Solonchaks (25,903.8 ha) [40]. A complex tidal amplitude is characterized by an uneven semidiurnal tide from the eastern sea and uneven diurnal tide from the western sea. The highest tide occurs commonly in October and November, and the lowest tide in April and May.

Both areas of accretion and erosion decreased in the period between 2001–2009 in the study area. The accretion area was reduced from 638.5 ha to 222 ha, and erosion areas were reduced from 371.6 ha to 355.6 ha. However, accretion and erosion increased during the period between 2009–2017 from 220.0 ha to 1097 ha and 355.6 ha to 496.0 ha, respectively [41]. The erosion process occurred only in the eastern area until the 20th century. However, the Gulf of Thailand area has been disturbed by erosion since 2001 [41]. Severe erosion has been detected at some points in the districts of Ngoc Hien, particularly Dat Mui commune, Phu Tan, and Tran Van Thoi, with an average sediment loss rate of ca. 0–15 m yr$^{-1}$. In contrast, strong accretion has been occurring in the districts of Nam Can and Ngoc Hien [41].

Natural mangrove forests in Ca Mau occupy 42,756 ha, accounting for 31% of the total mangrove area of Vietnam [42]. Nine species dominate mangrove forests, including *A. alba*, *A. marina*, *A. officinalis*, *B. parviflora*, *E. agallocha*, *L. racemosa*, *Phoenix paludosa*, *R. apiculata*, and *R. mucronata* in the Ca Mau peninsula, creating eight communities of pure or mixed species [43]. Since 2007 to the present, ca. 8870 ha of mangrove forests have been lost by estuarine and coastal erosion [44].

Seven study sites were selected: three accretion sites (site 1, 2, and 3) and four erosion sites (site 4, 5, 6, and 7) in the coastal districts of Nam Can (site 1, 2, 3, and 4), Ngoc Hien (site 5 and 7), and Phu Tan (site 6; Table 1; Figure 1). Con Trong Ong Trang and Con Ngoai Ong Trang are small islands formed by means of accretion processes at the Cua Lon Estuary during the 1960s and 1980s, respectively. With changes in mechanisms of the water currents and sediment movement within this estuary area, the northwestern banks where sites 1 and 2 are located have dramatically accreted seawards. In contrast, the southeastern banks have slightly eroded since 2004 (site 5) or 2014 (site 4). No evidence of erosion was found at site 6 until 1998. Site 3 is adjacent to site 7; however, sites 3 and 7 have opposite sediment drift. A sea-dam was established in 2011 that has prevented strong waves, reducing surface erosion at site 7. However, we detected many sand dunes extending offshore as a consequence of underground erosion processes at site 7. Soils at the study sites were classified as Gleyic Solonchaks [40]. Soil texture was silt loam or clay loam at accretion sites and clay at erosion sites.

### 2.2. Plot Establishment and Measurement of Trees, Saplings, and Seedlings

Three parallel transect lines with 100 m intervals from east to west were established from the shoreline to 150 m landwards at each study site. Impacts of coastal accretion and erosion processes were considered at coastal areas within 150 m from the sea.

Each transect included three 20 m × 20 m plots that were separated by 30 m along the transect at seaward (0–50 m), intermediate (50–100 m), and landward (100–150 m) zones. Two 20 m × 20 m plots were established for each transect line at sites 6 and 7 due to narrow coastal areas. Trees with a diameter at breast height (DBH) ≥ 6 cm were measured at a total of fifty-seven 20 m × 20 m plots. Three 2 m × 2 m subplots were established diagonally within a 400-m$^2$ plot. Some subplots were set to 1 m × 1 m or 10 m × 10 m depending on field conditions. Woody mangroves with DBH ≥ 6 cm were considered to belong to the canopy layer. Mangroves with a height ≥ 0.5 m and DBH < 6 cm were

considered saplings, and mangroves with a height < 0.5 m were considered seedlings. A total of 171 4-m$^2$ subplots were used for measuring saplings and seedlings.

**Table 1.** Topographic and coastal characteristics of the study sites.

| Site | 1 | 2 | 3 | 4 | 5 | 6 | 7 |
|---|---|---|---|---|---|---|---|
| Location | Seaward side of Con Ngoai Ong Trang Island: NW bank, Cua Lon Estuary | Seaward side of Con Trong Ong Trang Island: NW bank, Cua Lon Estuary | Dat Mui National Park, at the border between Gulf of Thailand and East Sea | SE bank of Con Ngoai Ong Trang Island, Cua Lon Estuary | SE bank of Con Trong Ong Trang Island, Cua Lon Estuary | Coastline of Gulf of Thailand, in front of sea dam | Dat Mui National Park, behind the sea dam |
| Accretion/erosion | Accretion | Accretion | Accretion | Erosion | Erosion | Erosion | Erosion |
| Formed | 1980s | 1960s | Before 1953 | Since 2014 | Before 2004 | After 1998 | 1940–1985 |
| Rate (m yr$^{-1}$) | 7.38 ± 7.61 (1992–2011) [45] | 48.69 ± 3.01 (1979–2011) [45] | 44.74 ± 26.76 (1953–2011) [45] | NA | NA | −40.8 (1998–2002) [46] | −10.28 ± 2.64 (1953–2011) [45] |
| Mean altitude (asl, m) | −1.9 | −2.0 | −3.0 | −1.8 | 2.2 | 1.3 | 1.3 |
| Soil texture | Silt loam or silt clay loam [39] | Silt loam or silt clay loam [39] | Silt loam or silt clay loam [39] | Clay | Clay | Clay | Clay |
| Soil salinity (‰) | 32.87 | 30.32 | 35.15 | 35.85 | 33.54 | 40.94 | 35.39 |

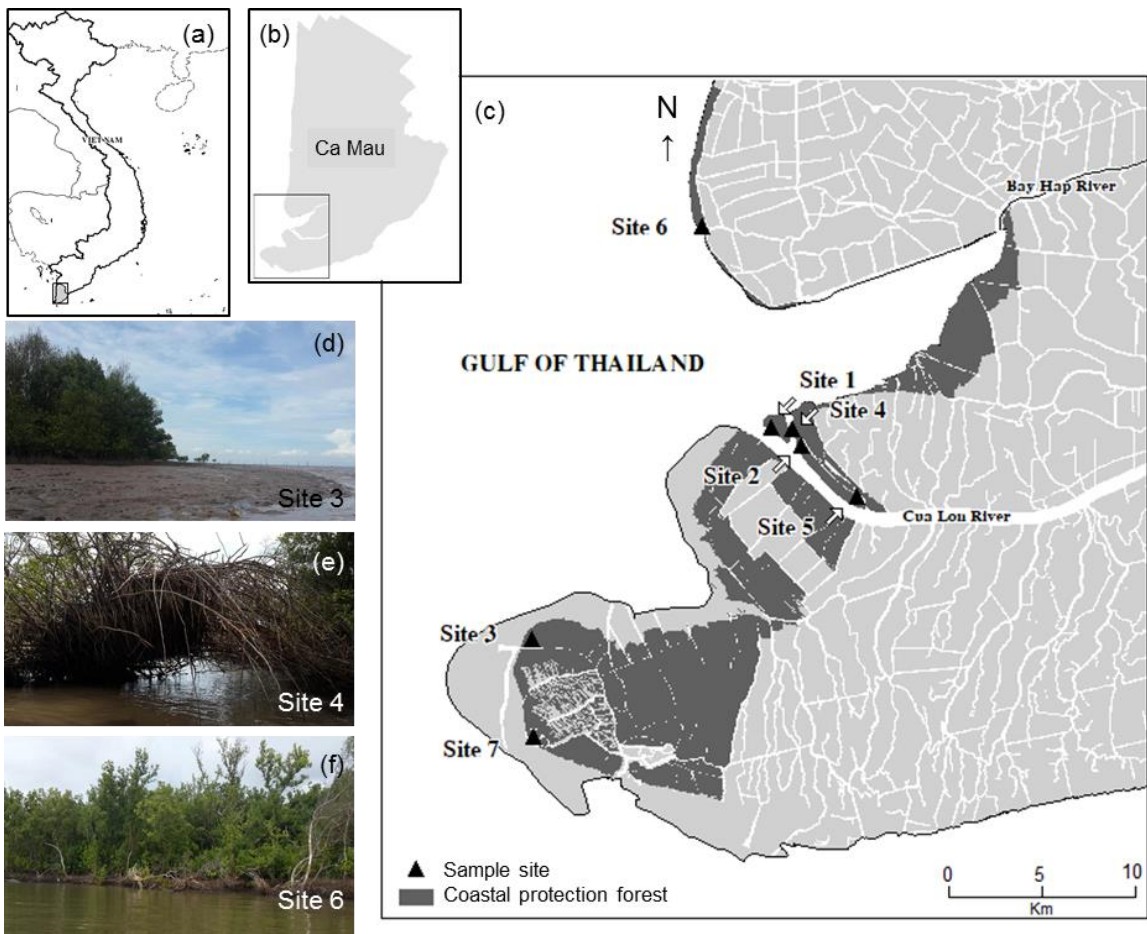

**Figure 1.** Map of (**a**) Vietnam, (**b**) Ca Mau province, and (**c**) the study site distribution as described in Table 1. Photos of (**d**) accretion and (**e**,**f**) erosion sites.

Species name, DBH, height, and XY coordinates were recorded for all trees with DBH ≥ 6 cm within a 20 m × 20 m plot. DBH was measured 1.3 m above the ground or at the upper point of the highest stilt root if root height exceeded 1.3 m [47]. Species name, height, and number of stems were measured for all seedlings and saplings within 4 m$^2$ subplots. Elevation and geographic coordinates of each plot were recorded using a GPS receiver (GPSMap 76CSx, Garmin, Olathe, KS, USA).

### 2.3. Data Analysis

DBH distribution by tree species for each plot was pre-examined to understand structural characteristics. Variables representing stand structural characteristics were selected for grouping plots based on the pre-examination (Table 2). *A. alba* and *R. apiculata* were the dominant species at accretion and erosion sites, respectively. The other species were scattered among the various plots. The size distribution of trees showed different characteristics depending on the distance from sea and the processes and degree of accretion or erosion. Thus, species composition and tree size were used as variables. The variables were number of species, relative densities of *A. alba* and *R. apiculata*, ratio of basal area of a species to total basal area in a plot, number of *A. alba*, *R. apiculata*, or other species ≥ 20 cm DBH, and number of *A. alba* < 10 cm.

**Table 2.** Structural variables used for grouping plots for accretion and erosion sites.

| Variable | Description |
|---|---|
| Number of species | Number of species in a plot |
| Relative density of *Avicennia alba* | $\frac{\text{Number of } A.\ alba}{\text{Total number of stems}}$ |
| Relative density of *Rhizophora apiculata* | $\frac{\text{Number of } R.\ apiculata}{\text{Total number of stems}}$ |
| Ratio of *A. alba* basal area to total stand basal area | $\frac{\text{Basal area of } A.\ alba}{\text{Total basal area of stems}}$ |
| Ratio of *R. apiculata* basal area to total stand basal area | $\frac{\text{Basal area of } R.\ apiculata}{\text{Total basal area of stems}}$ |
| Number of *A. alba* ≥ 20 cm DBH | Number of *A. alba* with DBH ≥ 20 cm |
| Number of *R. apiculata* ≥ 20 cm DBH | Number of *R. apiculata* with DBH ≥ 20 cm |
| Number of other species ≥ 20 cm DBH | Number of stems of tree species with DBH ≥ 20 cm except *A. alba* and *R. apiculata* |
| Number of *A. alba* < 10 cm DBH | Number of *A. alba* with DBH < 10 cm |

The Spearman's rank correlation was calculated between each variable to analyse the relationship among structural characteristics. The normality of data was tested using Shapiro–Wilk normality test, and some variables were not normally distributed.

Plots were grouped based on the similarity of certain structural features. The structural variables in Table 2 were standardized and used for the cluster analysis. Agglomerative hierarchical clustering with Ward method and Euclidean distance was conducted to group plots [48]. Accretion and erosion plots were treated separately. The distance level in the cluster analysis indicated the similarity of plots within and among groups. Variables in each group within accretion or erosion sites were compared using Kruskal–Wallis test. A Mann–Whitney *U* test was used for the comparison of each group. Significance was set at *p* < 0.05.

Species composition and DBH distribution of each group were examined to understand commonalities and differences of structural characteristics among each group. Species composition was examined using the importance value (IV, %) for each species, which is the sum of relative density, relative coverage, and relative frequency of a species [49]. Coverage was calculated using basal area at 1.3 m. Cluster analysis, correlation, and Kruskal–Wallis test were conducted using SPSS (v. 25, IBM SPSS Inc. Chicago, IL, USA, 2017).

## 3. Results

### 3.1. Grouping of Stands

The hierarchical cluster analysis grouped plots into four groups for accretion sites and three groups for erosion sites (Figure 2). For accretion sites, plots were classified into two groups as group A1 and the other three groups at distance level 25. Group A4 was divided from groups A2 and A3 at distance level 20 and A2 and A3 were divided at level 13. The cluster analysis classified E1 from E2 and E3 at the distance level 25, then divided E2 and E3 at level 13.

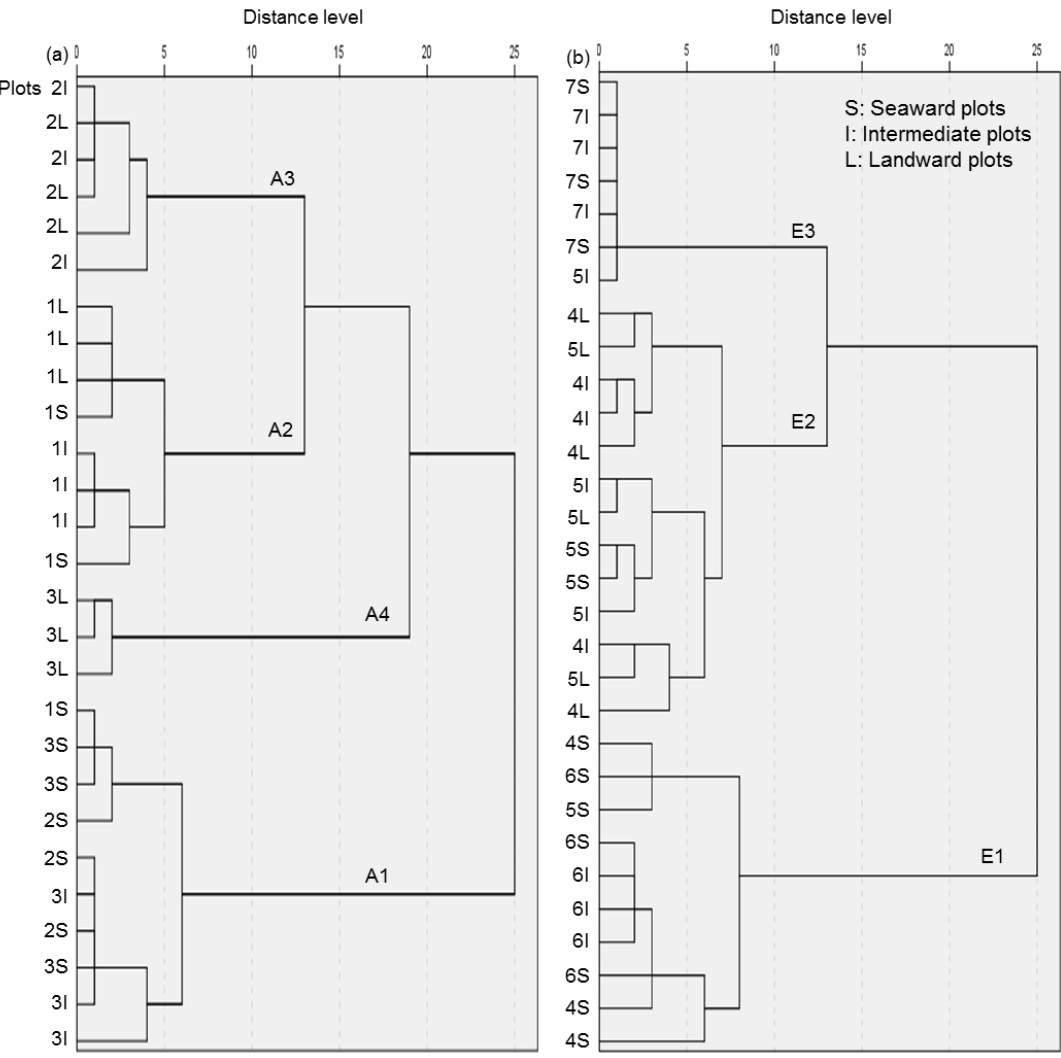

**Figure 2.** Hierarchical cluster analysis using Ward method, Euclidean distance, and Amalgamation steps as similarity measurements for (**a**) 27 plots in accretion sites (sites 1, 2 and 3) and (**b**) 30 plots in erosion sites (sites 4, 5, 6 and 7). Variables are shown in Table 2. Numbers in plots indicate site; Each site included 3 plots as seaward, intermediate and landward plots except in site 7; S indicates seaward plots, I intermediate, and L landward plots. A1, A2, A3, and A4 are groups for accretion sites and E1, E2, and E3 are groups for erosion sites.

Group A1 consisted of seaward plots. Intermediate plots of site 3 also belonged to A1. All plots in site 1 except the seaward plot belonged to A2. A3 consisted of intermediate and landward plots of site 2. Group A4 consisted of landward plots of site 3. E1 included all plots in site 6 and seaward plots of sites 4 and 5, where erosion processes were more recent than others (Table 1). All plots in sites 4 and 5 except two seaward plots (one from each site, 4 and 5, respectively) belonged to E2. E3

consisted solely of plots from site 7, which was characterized by a long period of erosive effects and recent underground erosion.

### 3.2. Structural Characteristics of Canopy Layer

#### 3.2.1. Species Composition in Each Group

Six species were distributed in the canopy layer over study sites (Figure 3). *A. alba* showed the highest importance value in groups A1, A2, and E1. A1 consisted of *A. alba* only. The importance value of *A. alba* was 59% in A2 and 36% in A3. *R. apiculata* overtook *A. alba* as the most dominant species in A3, with an importance value of 61%. *B. parviflora* had an importance value of 6% and 3% in A2 and A3, respectively. Two pioneer species, *A. alba* and *A. officinalis,* were distributed throughout A4, with an importance value of 42% and 58%, respectively.

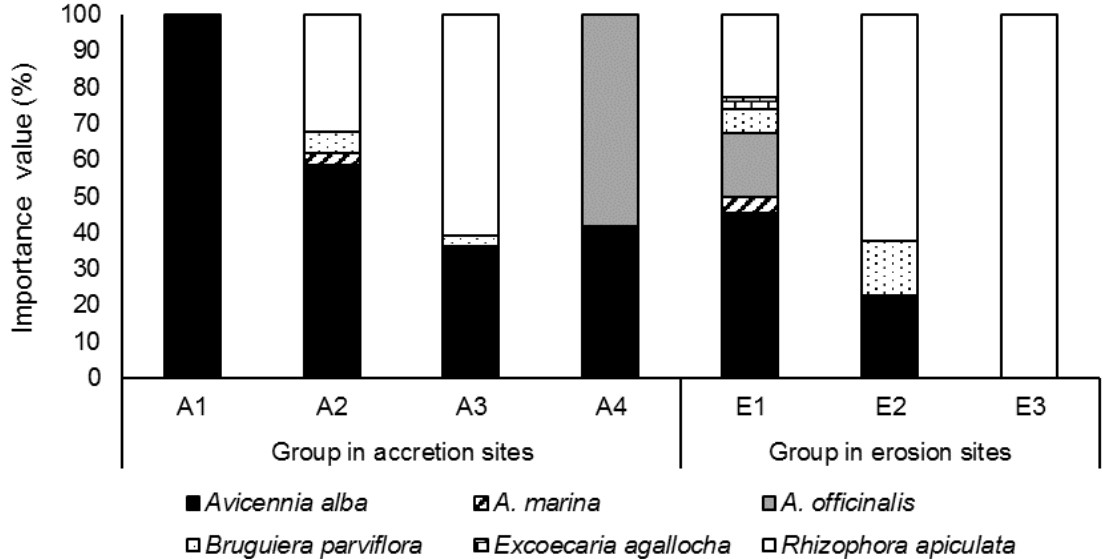

**Figure 3.** Importance value (%) of species in canopy layer for each group. A1, A2, A3, and A4 represent groups in accretion sites, and E1, E2, and E3 represent groups in erosion sites.

E1 had the highest number of species in the canopy layer. *A. alba* was the most dominant and abundant species in E1, followed by *R. apiculata*, and *A. officinalis*. *R. apiculata* was the dominant species in E2 and E3, with the highest importance value. E2 had only 3 species, and E3 consisted solely of *R. apiculata* in the canopy layer.

*A. alba* and *R. apiculata* shifted dominance among groups in both accretion and erosion sites. A Kruskal–Wallis test showed a significant difference in number of species, relative densities, and relative basal area of *A. alba* and *R. apiculata* among groups in both accretion and erosion sites ($p < 0.001$). A mean relative density of *R. alba* rank score was 25.1 for A1, 13.8 for A2, and 5 for A3 in accretion sites.

#### 3.2.2. DBH Distribution by Species in Accretion Sites

More than 90% of trees distributed in A1 were in the <20 cm DBH class, and few trees distributed in A1 and A2 were in the ≥35 cm DBH classes. DBH distribution of *A. alba* started at 10–15 cm DBH and 15–20 cm DBH in A3 and A4, respectively, and extended beyond 40 cm. The number of trees ≥ 40 cm DBH in A4 was much higher than that in A3.

*A. alba* showed a bell-shaped DBH distribution, whereas the DBH distribution of *R. apiculata* was reverse J-shaped in most groups (Figure 4). The peak and range of DBH distribution of *A. alba* increased from A1 through A2 to A3 in accretion sites. The peak DBH distribution of *A. alba* was 10–15 cm in A1 and A2, 20–25 cm in A3, and 25–30 cm in A4. *R. apiculata* was distributed up to 20–25 cm DBH and 35–40 cm DBH in A2 and A3, respectively, showing a wider DBH range and larger trees from A2 to

A3. *B. parviflora* and *A. marina* were under 10 cm DBH in A2 only. Only two species of *A. alba* and *A. officinalis* occurred in the canopy layer, with *A. alba* in larger DBH classes than *A. officinalis* in A4. *A. officinalis* was relatively evenly distributed from 10–15 to 25–30 cm DBH in A4.

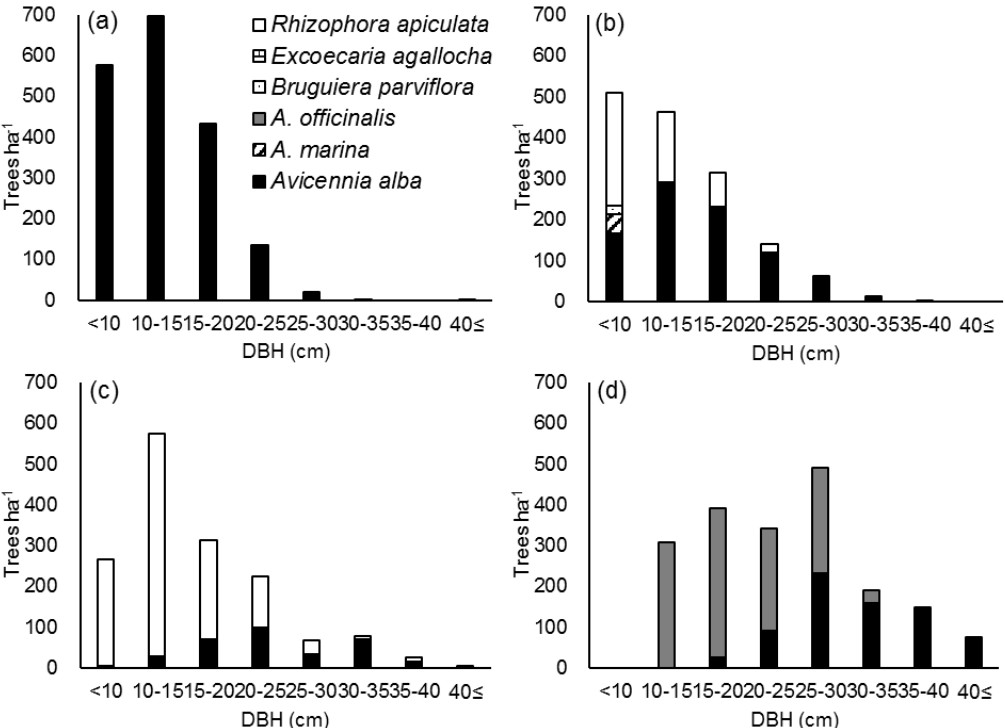

**Figure 4.** Diameter at breast height (DBH) distribution of trees ≥ 6 cm DBH by species for groups (**a**) A1, (**b**) A2, (**c**) A3, and (**d**) A4 in accretion sites. Plots belonging to each group are shown in Figure 2.

### 3.2.3. DBH Distribution by Species in Erosion Sites

E2 showed a much wider DBH distribution than the other groups, with trees found in all DBH classes (Figure 5). In contrast, most trees in E1 and E3 were found to be under 35 cm and 20 cm DBH, respectively. The distribution pattern of *A. alba* in erosion sites was similar to that in accretion sites. The *A. alba* distribution was up to 30–35 cm DBH, and the DBH distribution peaked at 10–15 cm DBH in E1. The peak and range of DBH distribution of *A. alba* increased from E1 to E2. *A. officinalis* was found only in E1 and showed a reverse J-shaped DBH distribution.

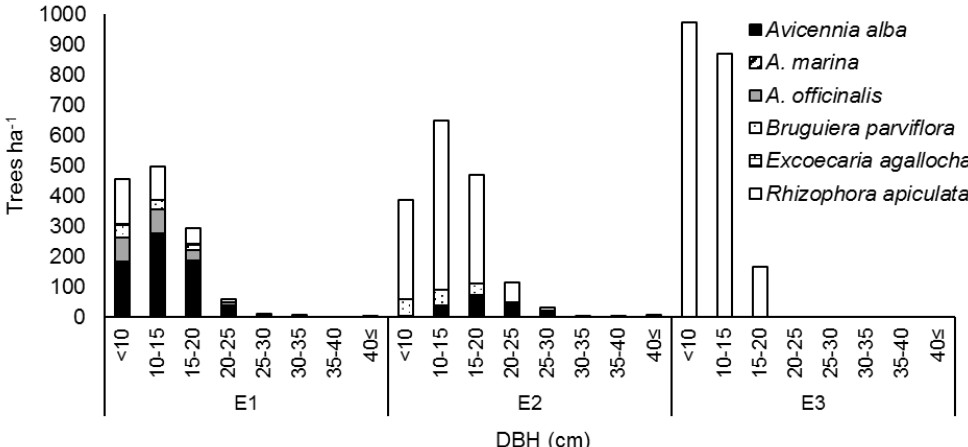

**Figure 5.** DBH distribution of trees ≥ 6 cm DBH by species for groups E1, E2, and E3 in erosion sites. Plots belonging to each group are shown in Figure 2.

The *R. apiculata* individuals were under 30 cm DBH range in E1 and E2. Further, *R. apiculata* showed a skewed bell-shaped DBH distribution, which peaked at 10–15 cm DBH in E2; it had a reverse J-shaped distribution in E1 and E3. Trees in E3 were less than 20 cm DBH. *B. parviflora* in E1 and E2 and the invading species, *E. agallocha*, in E1 were distributed in small DBH classes < 20 cm.

### 3.3. Regeneration on Accretion and Erosion Sites

Species occurrence remained the same between seedling and sapling layers in A1 and A2 (Figure 6). The occurrence of one or two species was different between seedling and sapling layers in A3 and A4. *B. sexangula* was observed in the sapling layer; however, it was absent in the seedling layer in A3. *R. apiculata* was found in both the seedling and sapling layers; however, *A. alba* and *B. cylindrica*, which appeared in the seedling layer, were missing in the sapling layer in A4. *A. officinalis* was found only in the sapling layer in A4. *S. alba* occurred in the seedling and sapling layers; however, they were not seen in the canopy layer in A1 and A2.

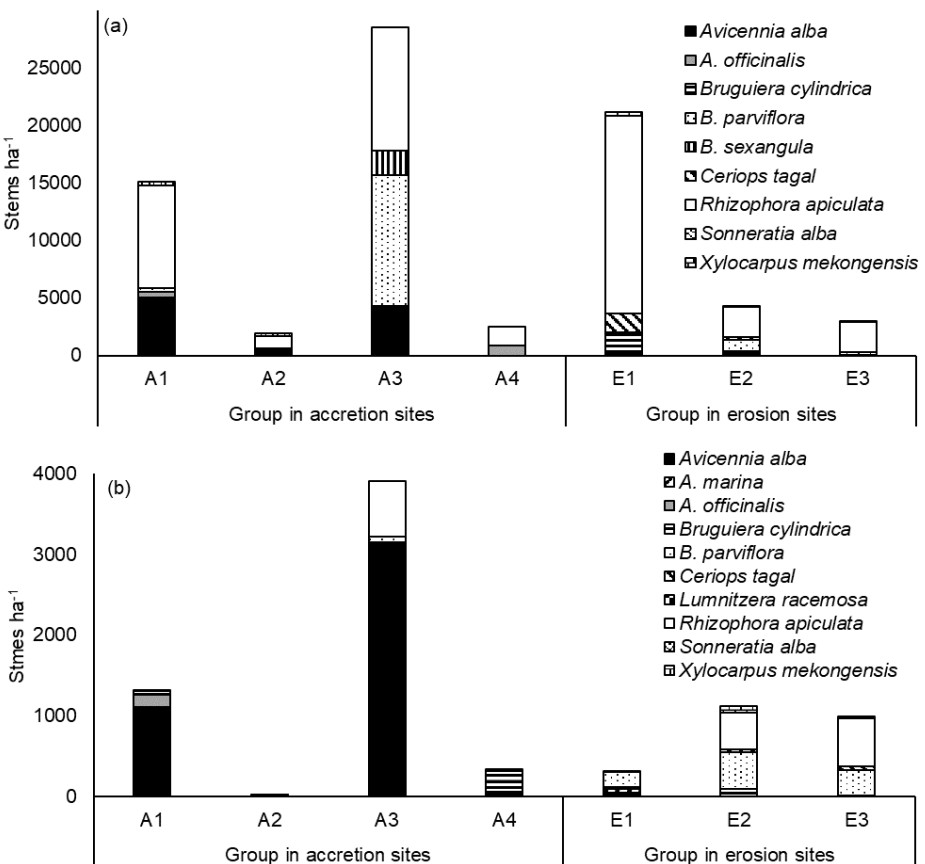

**Figure 6.** Density of (**a**) sapling (DBH < 6 cm and height ≥ 50 cm) and (**b**) seedling (height < 50 cm) by species in groups A1, A2, A3, and A4 in accretion sites and groups E1, E2, and E3 in erosion sites. Plots belonging to each group are shown in Figure 2.

Species occurrence between seedling and sapling layers showed a greater difference in erosion sites than in accretion sites. E1 had five species in both seedling and sapling layers; however, only *A. alba* was found in both layers, and the other species were found in either the seedling or sapling layers. Erosion sites had a greater number of species in the seedling and sapling layers than accretion sites. *C. tagal*, *L. racemosa*, and *X. mekongensis* occurred only in the erosion sites. *L. racemosa* occurred only in the seedling layer in all erosion sites.

*A. alba*, *B. parviflora*, and *R. apiculata* were the three most common species in the seedling and sapling layers in the study sites. However, their occurrence and abundance differed depending on

groups. *A. alba* was the most abundant species in the seedling layer in A1, A2, and A3. However, the ratio of *A. alba* to the overall species was less than 15% in both the seedling and sapling layers in A4 and all erosion sites. *R. apiculata* was the most abundant species in the sapling layers in erosion groups and A4, and in the seedling layer in E2 and E3. *B. parviflora* was distributed in both the seedling and sapling layers in all groups except A4 and E1.

### 3.4. Correlation of Structural Variables

The correlation between density and basal area of different species showed the potential for coexistence of some species and relatively mutual exclusion of others. *A. alba* and *R. apiculata* had a strong negative relationship in both relative density and relative basal area ($p < 0.05$; Table 3). Both densities of *A. alba* ≥ 20 cm DBH and <10 cm DBH had a negative relationship with relative density and relative basal area of *R. apiculata*. The number of other species ≥ 20 cm DBH showed a negative relationship with the number of *R. apiculata* ($p < 0.05$). Further, the number of species showed a positive correlation with relative basal area of other species and number of *R. apiculata* ≥ 20 cm DBH ($p < 0.05$).

**Table 3.** Spearman's correlation coefficients for the different variables attributed to grouping of plots in accretion and erosion sites. Variables represent number of species (# species), relative density (RD), and relative basal area (RBA) of tree species, number of trees ≥ 20 cm DBH (≥20), number of *A. alba* < 10 cm DBH (Aa < 10), and density (number of stems ha$^{-1}$).

| | # Species | RD Aa | RD Ra | RBAAa | RBARa | RBAOthers | Aa ≥ 20 | Ra ≥ 20 | Others ≥ 20 | Aa < 10 |
|---|---|---|---|---|---|---|---|---|---|---|
| RD Aa | −0.145 | | | | | | | | | |
| RD Ra | 0.069 | −0.916 ** | | | | | | | | |
| RBAAa | −0.153 | 0.980 ** | −0.854 ** | | | | | | | |
| RBARa | 0.099 | −0.936 ** | 0.987 ** | −0.888 ** | | | | | | |
| RBAOthers | 0.754 ** | −0.103 | −0.176 | −0.189 | −0.116 | | | | | |
| Aa ≥ 20 | 0.123 | 0.309 * | −0.286 * | 0.328 * | −0.349 ** | −0.022 | | | | |
| Ra ≥ 20 | 0.525 ** | −0.251 | 0.318 * | −0.211 | 0.278 * | 0.170 | 0.241 | | | |
| Others ≥ 20 | 0.251 | 0.026 | −0.266 * | −0.076 | −0.246 | 0.509 ** | 0.178 | 0.022 | | |
| Aa < 10 | −0.132 | 0.852 ** | −0.776 ** | 0.845 ** | −0.770 ** | −0.065 | −0.065 | −0.325 * | −0.071 | |
| Density | −0.019 | 0.205 | −0.213 | 0.212 | −0.223 | 0.080 | 0.012 | −0.264 | −0.101 | 0.211 |

**: $p < 0.01$, *: $p < 0.05$, D: density (number of stems ha$^{-1}$), Aa: *Avicennia alba*, Ra: *Rhizophora apiculata*, other species: species other than *A. alba* and *R. apiculata*.

## 4. Discussion

### 4.1. Effects of Accretion and Erosion on Mangrove Structure

Mangrove forests showed different patterns in species composition and stand structure depending on accretion or erosion processes where they were located. Mangrove development on accretion sites exhibited primary succession along the distance from the water [4,50]. Accretion processes created newly exposed mudbanks, providing sites for mangroves to enter. Seaward stands on erosion sites continuously experienced erosion, exhibiting gap dynamics at eroded sites [5].

The shift in dominance between *Avicennia* in accretion sites and *R. apiculata* in erosion sites could be explained by their species-specific characteristics and adaptation to coastal environments. *A. alba* is favoured over *R. apiculata* in areas of sediment accretion, relatively high disturbances or harsh environments, as well as higher intertidal locations [51]. *R. apiculata* is more sensitive to sediment burial than *A. alba* [52]. *A. alba* seedlings survived in up to 7 cm sediment burial [53], whereas the seedling mortality of *R. apiculata* increased linearly with increasing sediment accretion [52]. *A. alba* is a pioneer species and grows fast during the seedling stage, which enhances their ability to resist sediment disturbance [54,55]. Furthermore, *A. marina* and *A. officinalis* are tolerant to a wide range of soil salinity, enabling dominance at seaward sites [32]. *Avicennia* occupies seaward frontlines, with their combination of reproduction and vegetative regrowth, whereas *R. apiculata* dominates more stable environments such as landward sites [55,56].

Mangrove stands on accretion sites extended both seawards and landwards across intertidal zones. The distance from the shoreline reflected the timescale of accretion and stand development. Seaward

plots exhibited recent accretion, whereas floor substrates were more stabilized in landward plots on accretion sites. A1 consisted of seaward plots, A2 of intermediate, and A3 and A4 were landward plots. Changes in stand structure from A1 to A3 and A4 reflected stand development on accretion sites. Trees in larger DBH classes increased and stand density decreased from A1 through A3 and A4. Species and structural features in each group differed, reflecting the distance from the water and timescale of accretive processes. Mangrove forest structure was highly variable in canopy height, tree density, and tree basal area across the intertidal zone of a rapidly accreting mangrove site in New Zealand [57].

Erosion processes contribute to scattered disturbances, leading to gap dynamics and mosaic patterns of mangrove stands of diverse species and structure. Groups in erosion sites included a greater number of species than accretion sites, indicating that erosion sites might provide regeneration sites for a greater number of species than accretion sites [26]. Species capable of invading A1 might be limited, attributable to adaptations to newly created mudbanks that were still unconsolidated and had high soil salinity and inundation [58,59]. *C. tagal*, *E. agallocha*, *L. racemosa*, *X. mekongensis* reportedly adapt to a high firm floor or high tidal inundation position [32,43]. The distribution of *E. agallocha* in the canopy layer and *C. tagal*, *L. racemosa*, *B. cylindrica*, and *X. mekongensis* in the seedling and sapling layers in erosion sites suggests that seaward plots on erosion sites might provide more inland-like floors than accretion sites. E1, which consisted of seaward plots, had a greater number of species than E2 and E3, suggesting that E1 might have more frequent disturbance of erosion than E2 and E3, providing more space for the introduction of new species [51].

### 4.2. Development of Estuarian Mangrove Forests on Accretion Sites

Mangrove stands developed from a pioneer phase to a more mature phase based upon the distance from the water or timescale of the accretion processes [37]. A1 and A2 were dominated by *A. alba*, which is a representative pioneer species at the most foreshore in mangrove forests [60,61]. The seedling layer in A3 was still dominated by *A. alba*, and the canopy layer in A4 was dominated by *A. officinalis*, which is also a pioneer species. However, A3 and A4 included larger trees than A1 and A2, as well as more landward species in the understory layer [51], suggesting more development than A1 and A2.

Stand development in mangroves occurs in four stages: colonization—early development—maturity—senescence; each stage includes different structural characteristics in terms of density and biomass of trees [51]. Stand density changes from low to high at colonization, very high at early development, medium at maturity and low at senescence [59]. Accretion groups exhibited similar development patterns. A1 could be considered to be in the colonization of pioneer species stage, A2 in the early development stage, and A3 and A4 in a more mature stage.

A1 was characterized by the full dominance of *A. alba* in the canopy layer and the majority of trees were under 20 cm DBH, indicating the stands were in the pioneer phase. The density of *A. alba* was highest in the seedling layer; however, the sapling layer was dominated by *R. apiculata* in A1. *A. officinalis*, *R. apiculata*, *B. parviflora*, and *S. alba*, which were absent in the canopy layer, were distributed in both the seedling and sapling layers in A1. *S. alba* grows at the forest edge, particularly in front of *A. alba*, playing an important role in establishing appropriate habitats for the colonization of *A. alba* [26,62]. Therefore, the occurrence of *S. alba* in both the seedling and sapling layers in A1 and A2 indicated seaward development of mangroves even though the density was limited.

We considered A2 to be in the transition phase from A1 to A3. *R. apiculata* and *B. parviflora* occurred in the canopy layer. The most dominant species shifted from *A. alba* in A2 to *R. apiculata* in A3. The peak and DBH range of *A. alba* and *R. apiculata* also increased to larger DBH classes from A2 to A3. *A. alba* and *A. officinalis* dominated the canopy layer, with *A. alba* in larger DBH classes only and *A. officinalis* overtaking lower DBH classes in A4. However, *B. cylindrica* is a newly invading species, with the highest density in the seedling layer; *R. apiculata* was the most abundant species in the sapling layer in A4, implying that the stand would be dominated by *Rhizophora* later, leading to a structure similar to A3 [63]. Changes in dominant species from *Avicennia* to *Rhizophora* along

fringe–basin–inland transition forests with an increasing size in diameter, height, and basal area were also reported in mangrove forests in Guaratiba, Brazil [36].

### 4.3. Mosaic Pattern of Mangrove Forests on Erosion Sites

Mangroves are resilient to various types of disturbances; however, their recovery depends upon the frequency, intensity, size, and duration of disturbances [64]. Erosion processes work as small scale disturbances and lead to a mosaic of various mangrove stands over time [5]. E1, E2, and E3 showed variations in stand structure, reflecting temporal differences in erosion, erosion rate, and distance from the water.

Both E1 and A1 were located at the water edge. A1 was solely dominated by *A. alba*, providing an example of the pioneer stage, whereas E1 included six species in the canopy layer, including *B. parviflora* and *R. apiculata*, which are shown in A2 and A3. The pioneer stage was often missing in coastal erosion areas [37]. The overlapping species between E1 in erosion sites and A2 and A3, which included intermediate or landward plots in accretion sites, indicated that E1 might provide more landward environments than A1. *E. agallocha*, *C. tagal*, *L. racemosa*, and *X. mekongensis* were found only in erosion sites. *A. marina*, *C. tagal*, *L. racemosa*, *Bruguiera*, *Rhizophora*, and *Xylocarpus* were found in the seedling or sapling layers of E1 and were reported on the landward zone in Portuguese Island, Mozambique [59]. Another study in King Sound, north-western Australia also showed the landward distribution of *Bruguiera*, *C. tagal*, *E. agallocha*, and *Xylocarpus* in coastal mangrove forests [25].

E1 had the highest number of species in the overstory and understory. The number of species in seedling and sapling layers were five and five, respectively, which was similar or lower than that in E2 and E3. However, only *A. alba* simultaneously occurred in both seedling and sapling layers, whereas the other species occurred in either one of the two layers, implying that seedlings had difficulty in growing up to sapling layers, attributable to frequent erosion disturbances at the seaward frontline in erosion sites [54,65].

E2 was a mixture of intermediate or landward plots in sites 4 and plots in site 5, reflecting weaker effects of erosion and a more interior environment, such as lower soil salinity than that of E1 [32]. E1 included site 6, where the erosion rate is $-40.8$ m yr$^{-1}$, which was a much higher rate than that of $-10.28$ m yr$^{-1}$ in site 7 (E3). E2 and E3 had fewer species than E1 in both the overstory and understory. The relative density of *R. apiculata* increased from E2 to E3. Site 7 of E3 was under erosion the longest among the study sites; however, the erosion rate was lower than that of E2. E3 consisted of plots in site 7 which were located behind the sea-dam and only underground erosion occurred. The establishment of the sea-dam in E3 might provide more landward environment by reducing erosion and tidal impact since 2011, resulted in the dominance of *R. apiculata* in the canopy layer. The differences in species composition and mangrove structure between E1 and E3 implied the effects of dam construction on mangrove structure, and the potential for changes in mangrove ecosystems. Variations in species composition and stand structure between E1 and E2 or E3 reflected erosion rate and timescale of erosion, resulting in a mosaic pattern of stands over mangrove forests [5].

## 5. Conclusions

Species composition and stand structure differed between accretion and erosion sites. Changes in species composition and stand structure reflected the distance from the shoreline and accretion and erosion processes. *A. alba* and *R. apiculata* were representative species in accretion and erosion sites, respectively. The dominance of *A. alba* shifted to *R. apiculata* as stands developed landward. Mangrove stands showed structural development from the pioneer to mature stage from seaward to landward in accretion sites. Seaward erosion sites exhibited small scale disturbances, providing openings for species to invade, resulting in mosaic patterns of stands over mangrove forests affected by timescale and rate of erosion. The differences in species composition between accretion and erosion sites or seaward and landward plots indicated that species selection should be carefully considered in mangrove restoration projects. The mangrove structure in E3 showed that sea-dam construction

facilitated changes in mangrove development from seaward to landward, which would affect habitats where coastal communities depend on mangroves. Mangroves have adapted to coastal accretion and erosion processes, while recent acceleration of erosion may disrupt mangrove maintenance in coastal areas. An improved understanding of mangrove dynamics on coastal edges is urgent to conserve and restore mangrove ecosystems.

**Author Contributions:** L.T.M.N. and P.S.P. conceived and designed the study. L.T.M.N. and H.T.H. collected the data. L.T.M.N. and P.S.P. analyzed the data and wrote the manuscript. H.V.T. collected information on study sites and produced maps. All the authors discussed the results and reviewed and approved the manuscript. All authors have read and agreed to the published version of the manuscript.

**Funding:** This research was supported by the Korea Forestry Promotion Institute Scholarship Program, and R&D Program for Forest Science Technology (Project No. 2013069C10-1919-AA03) of the Korea Forestry Promotion Institute, Korea Forest Service.

**Acknowledgments:** We thank Khiem, H.T. and technical staffs in South Western Forest Research and Experimental Center for data collection in the field. We acknowledge the Research Institute of Agriculture and Life Sciences, Seoul National University, for language assistance. We are very grateful to the members of the Forest Ecology and Silviculture Laboratory, Seoul National University and the anonymous reviewers for their valuable comments.

**Conflicts of Interest:** The authors declare no conflict of interest.

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
