# Peer review of "Comparison of Mangrove Stand Development on Accretion and Erosion Sites in Ca Mau, Vietnam"

_forests, doi:10.3390/f11060615_

Round 1
Reviewer 1 Report
The future of mangrove attracts global attention because of their importance in the ecological and physical dynamics of coastal systems in a context of global environmental changes and sea-level rise. This study of mangrove evolution makes sense to their protection.
This paper analyzes the composition/structuration of mangrove stands in accretion or erosion situation based on field measurements. The title and abstract are appropriate for the content of the text. Furthermore, the article is well constructed, the experiments were well conducted, and analysis seem to be well performed. Overall the paper is well written and presented. The cartography within this paper is clear but the quality of the graphs and tables can be improved.
The manuscript is certainly very interesting, well conducted and of general relevance. I think the authors should consider and discuss of the evolution mangroves stands and its impacts on the sediment dynamics and biogeomorphic feedbacks between abiotic and biotic factors in such mangrove environments. Also, review of linking the observed change to natural and anthropogenic historic processes, could be shown or can be discussed somewhere.
Others comments
- Bibliographic references are sometimes too old (for example references 1, 3, 10, 11…)
More recent and relevant references must be cited int your introduction.
- Page 52 – Authors say “However, few studies of mangal development have been conducted on naturally-retreating coasts”, I am not agree this this statement. Several studies have been conducted on this theme around the world (some examples):
Along the coast of Guianas:
- Fromard et al, Half a century of dynamic coastal change affecting mangrove shorelines of French Guiana. A case study based on remote sensing data analyses and field surveys
- Plaziat et Augustinus, 2004. Evolution of progradation/erosion along the French Guiana mangrove coast: a comparison of mapped shorelines since the 18th century with Holocene data
Along the coast of Thailand:
- Thampanya et al, 2006. Coastal erosion and mangrove progradation of Southern Thailand
Photographs of the various mangrove structures would be useful to visualize both accretion and erosion situations found.
Author Response
May 24, 2020
Responses to Reviewer 1 Comments
Manuscript ID: Forests-802322
Please find below point-by-point responses to the reviewer comments on our revised manuscript submitted to MDPI forests (Manuscript ID: forests-802322). We have listed the reviewers’ comments and inserted our responses in boxes beneath each section of comments. The line numbers we put at the end of each answer are line numbers in the revised version with track change. We are very grateful for all the remarks made by reviewers.
Response to Reviewer 1
The cartography within this paper is clear but the quality of the graphs and tables can be improved.
|
We modified table 1 and figures 1, 2, 3 and 6 and captions for the table and figures. |
I think the authors should consider and discuss of the evolution mangroves stands and its impacts on the sediment dynamics and biogeomorphic feedbacks between abiotic and biotic factors in such mangrove environments. Also, review of linking the observed change to natural and anthropogenic historic processes, could be shown or can be discussed somewhere.
|
We modified the sentence to reflect the comments. - The paragraph on the interaction of accretion/erosion/sediment dynamics and mangroves were modified and the impact of sediment accretion on mangroves were added. L 48-52. - The observed change by anthropogenic activities or dam construction were added in introduction and discussion. L 56-59, L421-426. |
Others comments
Bibliographic references are sometimes too old (for example references 1, 3, 10, 11…)
More recent and relevant references must be cited int your introduction.
|
We replaced references if there were more recent references. The reference changes were not indicated by track change because they were too many. |
L51-52 – Authors say “However, few studies of mangal development have been conducted on naturally-retreating coasts”, I am not agree this this statement. Several studies have been conducted on this theme around the world (some examples):
Along the coast of Guianas:
Fromard et al, Half a century of dynamic coastal change affecting mangrove shorelines of French Guiana. A case study based on remote sensing data analyses and field surveys
Plaziat et Augustinus, 2004. Evolution of progradation/erosion along the French Guiana mangrove coast: a comparison of mapped shorelines since the 18th century with Holocene data
Along the coast of Thailand:
Thampanya et al, 2006. Coastal erosion and mangrove progradation of Southern Thailand
|
- Thank you for catching this. The sentence was modified saying that the information is still limited. L 62-63. - Thank you for including suggested literature that we had missed. We added suggested references. References [10], [27], and [37]. |
Photographs of the various mangrove structures would be useful to visualize both accretion and erosion situations found.
|
- Representative photos for accretion and erosion sites were added in Fig. 1. |

Reviewer 2 Report
This manuscript compare the mangrove plant communities between sites with different degree of accretion and erosion in Vietnam. The content of this manuscript can be accepted after major revision. My specific comments are as below and must be addressed all:
Line 110-112, the authors stated “Seven study sites were selected and designated as three accretion sites (site 1, 2, and 3) and four erosion sites (site 4, 5, 6, and 7) in the coastal districts of Nam Can (site 1, 2, 3, and 4), Ngoc Hien (site 5 and 7), and Phu Tan (site 6; Fig. 1).” What is the criteria to designate as accretion site or erosion site. Any indication or physical properties you are using to classify these two groups of sites?
Table 1. Since the sites are located in different region of the river, I think the salinity of the soil water are also different among sites. Any data on this to supplement into table 1?
Fig. 1. The names has many error. The name should be Gulf of Thailand not Thailand Gulf. East Sea is not correct, I think it should be South China Sea. Why the colour of Sea is white in colour. It make the map looks very confusing. I suggest the authors should make a better map or use google earth as the base map and acknowledge google earth in figure legend for the illustration. Also I suspect the name “commune” may not be correct Name of major river should be indicated in the map. Really, the quality of this map is under the acceptance standard.
Figure 2, what is 4S, 4L etc. I can guess number is site number, but what is S and L. It should be stated in detail in figure legends
Figure 3 contains very big error if I am correct. In methods, you said three accretion sites (site 1, 2, and 3) and four erosion sites (site 4, 5, 6, and 7). But here, there are 3 erosion site and 4 deposition site. Why methods say 4 erosion site but there are only three erosion site. Why there is suddenly deposition site? Where are the accretion sites then? Are they the same?
DBH should be explained in the figure legend, although you only mentioned once in the text about the full name.
The discussion only concern about the mangrove plant variations. The last paragraph of the discussion should mention the importance of macroinvertebrates among mangrove with different ages/erosion. In such extreme sites in your case, macroinvertebrates is also an very important factor to affect the soil properties. Although the authors has not included this in the sampling, the authors should mentioned the variation of macroinvertebrates between these sites can be also important variables. I would suggest author cite the following reference and mention the importance of mancoinvertebrates:
Li YF, Du FY, Gu YG, Ning JJ, Wang LG. 2017. Changes of the macrobenthic faunal community with stand age of a non-native mangrove species in Futian Mangrove National Nature Reserve, Guangdong, China. Zool Stud 56:19. doi:10.6620/ZS.2017.56-19.
Crabs can also make burrows along the mangrove with strong salinity gradient and affect the physical components of the soils. The author should also mention such importance of crabs and its distribution according to salinity variations. This can also happen in your site although this has not been addressed. I suggest to read and cite the reference below:
Theuerkauff D, Rivera-Ingraham GA, Roques JAC, Azzopardi L, Bertini M, Lejeune M, Farcy E, Lignot J, Sucré E. 2018. Salinity variation in a mangrove ecosystem: a physiological investigation to assess potential consequences of salinity disturbances on mangrove crabs. Zool Stud 57:36. doi:10.6620/ZS.2018.57-36
Author Response
May 24, 2020
Responses to Reviewer 2 Comments
Manuscript ID: Forests-802322
Please find below point-by-point responses to the reviewer comments on our revised manuscript submitted to MDPI forests (Manuscript ID: forests-802322). We have listed the reviewers’ comments and inserted our responses in boxes beneath each section of comments. The line numbers we put at the end of each answer are line numbers in the revised version with track change. We are very grateful for all the remarks made by reviewers.
Response to Reviewer 2
Line 110-112, the authors stated “Seven study sites were selected and designated as three accretion sites (site 1, 2, and 3) and four erosion sites (site 4, 5, 6, and 7) in the coastal districts of Nam Can (site 1, 2, 3, and 4), Ngoc Hien (site 5 and 7), and Phu Tan (site 6; Fig. 1).” What is the criteria to designate as accretion site or erosion site. Any indication or physical properties you are using to classify these two groups of sites?
|
Thank you for catching this. It was typo. We used wrong terminology “designate”. We deleted “designate” and modified the sentence. - We selected the sites based on accretion or erosion processes. Table 1 explains the details. |
Table 1. Since the sites are located in different region of the river, I think the salinity of the soil water are also different among sites. Any data on this to supplement into table 1?
|
- We added soil salinity information in table 1. |
Fig. 1. The names has many error. The name should be Gulf of Thailand not Thailand Gulf. East Sea is not correct, I think it should be South China Sea. Why the colour of Sea is white in colour. It make the map looks very confusing. I suggest the authors should make a better map or use google earth as the base map and acknowledge google earth in figure legend for the illustration. Also I suspect the name “commune” may not be correct Name of major river should be indicated in the map. Really, the quality of this map is under the acceptance standard.
|
Thank you. We took to heart the suggestions that we had missed. We modified Figure 1. - The names were modified to Gulf of Thailand - Commune was an administrative unit. Commune was deleted. - The name of major rivers (Bay Hap River and Cua Lon River) were added in Figure 1. |
Figure 2, what is 4S, 4L etc. I can guess number is site number, but what is S and L. It should be stated in detail in figure legends
|
Thank you very much for catching this and sorry for confusing. We modified the Figure 2. - Title for Axis (distance level) was added. - A represents groups in “A”ccretion sites. D1, D2, .. were replaced as A1, A2, …, respectively. - We added more explanation for S and L in figure legends. |
Figure 3 contains very big error if I am correct. In methods, you said three accretion sites (site 1, 2, and 3) and four erosion sites (site 4, 5, 6, and 7). But here, there are 3 erosion site and 4 deposition site. Why methods say 4 erosion site but there are only three erosion site. Why there is suddenly deposition site? Where are the accretion sites then? Are they the same?
|
Thank you very much for catching this. It was typo and sorry for confusing. Deposition should be accretion. - Three accretion sites and four erosion sites were correct. By hierarchical clustering, plots in were grouped to A1, A2, A3, and A4 for deposition sites, and E1, E2 and E3 for erosion sites. - We modified axis title and added figure explanation in Figure 3. Deposition was changed to Accretion. D1, D2, D3 and D4 were replaced by A1, A2, A3 and A4, respectively. |
DBH should be explained in the figure legend, although you only mentioned once in the text about the full name.
|
“Diameter at breast height” was added to figure caption: Figures 4 and 5. |
The discussion only concern about the mangrove plant variations. The last paragraph of the discussion should mention the importance of macroinvertebrates among mangrove with different ages/erosion. In such extreme sites in your case, macroinvertebrates is also an very important factor to affect the soil properties. Although the authors has not included this in the sampling, the authors should mentioned the variation of macroinvertebrates between these sites can be also important variables. I would suggest author cite the following reference and mention the importance of mancoinvertebrates:
Li YF, Du FY, Gu YG, Ning JJ, Wang LG. 2017. Changes of the macrobenthic faunal community with stand age of a non-native mangrove species in Futian Mangrove National Nature Reserve, Guangdong, China. Zool Stud 56:19. doi:10.6620/ZS.2017.56-19.
|
Yes, macroinvertebrates are an important factor to affect soil properties in coastal ecosystems. However, it was difficult for us to discuss macroinvertebrates communities and mangrove structure or soil properties because we did not have data for that. Instead, we modified sentences, trying to consider coastal fauna/communities living in mangroves in introduction, discussion and conclusion. L 36-38, 420-421, 434-436. Reference was added [3]. |
Crabs can also make burrows along the mangrove with strong salinity gradient and affect the physical components of the soils. The author should also mention such importance of crabs and its distribution according to salinity variations. This can also happen in your site although this has not been addressed. I suggest to read and cite the reference below:
Theuerkauff D, Rivera-Ingraham GA, Roques JAC, Azzopardi L, Bertini M, Lejeune M, Farcy E, Lignot J, Sucré E. 2018. Salinity variation in a mangrove ecosystem: a physiological investigation to assess potential consequences of salinity disturbances on mangrove crabs. Zool Stud 57:36. doi:10.6620/ZS.2018.57-36
|
Burrows by crabs are an important factor on coastal soil properties. However, it was difficult for us to use the reference on salinity variations and mangrove crab distribution because our soil salinity data were not enough to discuss coastal communities. We added reference on the role of burrows by crabs on mangrove carbon sink [2]. |

Reviewer 3 Report
Dear authors,
You paper has a significant potential. I included all my observation highlighted with comments in the pdf file.
In general, the paper is very confused to read, because, in what you say in the results as group A1 and etc are different from what is in the figure 2. For this reason, I could not follow your results as I should. It is important to keep consistency along your text.
I would like to see where these groups are in the spatial context. They are part of which site sample? In figure 1 you show the samples sites, but after you do not use this sites locations in a final resulted map?
All in all, I hope that my comments can help you to improve your manuscript.
Best regards

Author Response
May 24, 2020
Responses to Reviewer 3 Comments
Manuscript ID: Forests-802322
Please find below point-by-point responses to the reviewer comments on our revised manuscript submitted to MDPI forests (Manuscript ID: forests-802322). We have listed the reviewers’ comments and inserted our responses in boxes beneath each section of comments. The line numbers we put in front of the reviewer’s comments are line numbers in the first MS and that at the end of each answer are line numbers in the revised version with track change. We are very grateful for all the remarks made by reviewers.
Reviewer 3:
In general, the paper is very confused to read, because, in what you say in the results as group A1 and etc are different from what is in the figure 2. For this reason, I could not follow your results as I should. It is important to keep consistency along your text. I would like to see where these groups are in the spatial context. They are part of which site sample? In figure 1 you show the samples sites, but after you do not use this sites locations in a final resulted map?
|
Thank you very much for catching this critical error and sorry for confusing. We modified the Figure 2 and relevant contexts. - “A” represents groups in “A”ccretion sites. D1, D2, .. were replaced as A1, A2, …, respectively. - We have 7 study sites: 3 sites in accretion area, and 4 sites in erosion area. Each site included 3 plots. We did cluster analysis using all plots for each accretion and erosion site, and regrouped plots based on the cluster analysis. We named the groups as A1, A2, for accretion sites and E1, E2 and E3 for erosion sites. Plots that belong to each group are shown in Figure 2. We added figure legends so readers can understand better. - The locations of sites are shown in Figure 1. Plots belonging to each group are shown in Figure 2. |
L2: what do you mean? structure? species composition?
|
Stand development refers to the changes in a stand over time, thus includes changes in both stand structure and species composition (Olive and Larson 1996). We investigated changes in mangrove species and structural characteristics, thus we used stand development. - Oliver, C.D.; Larson, B.C. Forest Stand Dynamics; John Wiley and Sons: New York, USA, 1996. |
L13. Maybe is better to be more specific here. Mangrove ecosystems are adapted...
|
- We meant that “mangrove species and mangrove forests” were adapted to coastal processes. Our study investigated changes in species composition and structural characteristics in mangrove forests, however, did not much investigate functions of mangrove forests. Thus, we used Mangroves. - Mangroves collectively represent shrubs and trees that belong to the families Rhizophoraceae, Acanthaceae, Lythraceae, Combretaceae, and Arecaceae; that grow in dense thickets or forests along tidal estuaries, in salt marshes, and on muddy coasts. Mangrove also refers to thickets and forests of such plants. - Our study partly focuses on changes in dominant species in mangrove forests along the distance from the water. |
L15-16. Where is your objective? Do your have scientific questions? Why it is important to make this study?
|
- The study objectives were “to provide insight into mangrove forest development in response to shoreline accretion and erosion processes”. L15-16. - The objectives are also shown in L 88-92. - Scientific questions: We thought species composition and structural characteristics would be different between accretion and erosion sites. - Conservation efforts for mangroves such as mangrove restoration are actively conducted over the tropics and subtropics. Appropriate species selection and target structure are the key for successful mangrove management. Our results will contribute to mangrove conservation and management practicies. |
L26 Can you explain why or how this happen? Timescale of erosion and erosion rate affected species composition and regeneration on erosion sites.
|
- More species invaded sites with a higher erosion rate or that were more recently eroded. Therefore, higher erosion rate resulted in more species invasion. Newly eroded sites had more number of species, implying that time of erosion affected species composition and regeneration on erosion sites. - We modified the sentence. |
L27 Invaded or colonized? be sure about the right term here.
|
- Modified to “colonized”. L27 |
L28 Reviewer 3: Ok. you said that in the beginning, however, how different is the answer here. Give some examples of the adaptations in each category (accretion and erosion).
|
- Accretion led to newly exposed bare soil or siltation, resulting in dominant species adapted to rapid colonization or siltation that was A. alba. Mangrove stands developed from dense A. alba dominant to R. apiculata dominant stands with increasing tree size at accretion sites. Thus, dominant species shift from A. alba to R. apiculate with bell shaped DBH distribution to reverse J shaped DBH distribution at accretion sites. - More species colonized sites with a higher erosion rate or that were more recently eroded at erosion sites. |
L33 Better start with a general concept of mangroves with the distribution and etc.
|
- We added the global distribution of mangrove forests. - We modified the sentence. L35-38. |
L40 geomorphological? maybe your term is more appropriate. But it is more common the geomorphological processes
|
- Changed. L45. |
L41 Suspended sediment? Suggestion: Maybe could be nice to have the definition of sediment. For example: https://www.usgs.gov/special-topic/water-science-school/science/sediment-and-suspended-sediment?
|
- changed. L46. |
L56-57. this part is a bit confused. How can you use NDVI for establishment of young mangrove seedlings...?
Bette to rewrite.
|
Sorry for confusing. We deleted “seedling” part. L66-69. |
L62 granulometry? sediment factor seems a bit stange. Be sure about this term.
|
- changed to sediment characteristics. L74. |
L62 their impacts? of what? Colonization of mangroves?? This sentence is too long and confused.
|
- We deleted the sentence. |
L89-91 Is there a reference for this
|
- Reference was added. L103. |
L92 s this area considering the total area in the Ca Mau? or regarding your study area?
|
- The area covers the total Ca Mau province. We modified the sentence. L103. |
L123 Table 1. Put the table in one page. Also, do you have the species composition at each site? It could be a nice information.
|
- We modified the arrangement of the table. - Information on species composition at study area was mentioned in L 119-121. - Species composition at study sites was the major part of the results. L 214-295. |
Table 1. Where did you find this classification of very high, high and low of accretion or erosion?
|
Accretion or erosion is facts on the ground. The degree of accretion or erosion (high or low) was determined based on the rate and time scale of accretion/erosion (shown in Table 1) and personal communication with local authorities. |
Table 1. Provide the complete reference. There are many books and reports from FAO. Be careful with the formate of this table. put the name Solonchaks in one line. Also, if all sites have the same soil type you can write it in the text and does not need to put in the table.
|
Soil type was deleted because it was explained in the text. L 103-105. |
Figure 1. Please, give more information. For example, tell about the small figure in the above left side. Also, again tell that it is in Ca Mau, Vietman and etc. The coordinates are difficult to see in the map.
|
Figure 1 was modified and explanations were added. |
L144-145. This seems not the right way to make the reference. Olathe, Kanksas, USA it is not necessary here. It can make a confusion for the reader.
|
Corrected. L159. |
L145-146. What? Do you think it is necessary to say?
|
Deleted. |
Table 2. It is a bit strange to see a table with just one column. Suggestion: Make bullets instead of a table. Or give more information for each variable. you could include another column with the equation of the relative density.
what do you mean? number of individuals or number of species?You can have 4 tree species extra A. alba and R. apiculata. Each of these 4 species can have more than one individual in each plot. So, give your matematical approach more clear here and in other variable, for instance, relative density.
|
- A column was added to describe the variables. Equations were added. |
L173-174. check if this is the right way to cite. I think you do not need all informaiton here.
|
Modified. L188. |
165-166. Give more information here. Is the distance level a index? There is a equation of this distance level? Do you know if this distance level varies between wich range? What is low and high for this distance level?
|
- Distance level is not an index. Lower distance level indicates the two groups are similar, while greater distance level means the two groups are different. - There are various algorithms and parameters (Euclidian distance vs Mehalanobis distance vs Minkowski distance, single linkage vs complete linkage vs.., number of expected clusters, …). Users choose algorithm and parameters based on their dataset. We used Agglomerative hierarchical clustering with Ward method and Euclidean distance, because our plot numbers are 27 and 30 in accretion and erosion sites, respectively, that were not many. - Distance level in cluster analysis can vary depending on algorithm and parameters. We interpret the similarity of groups by comparing its distance level at its grouping. - We changed the term homogeneity to similarity and included reference [48]. |
L179. what this number means? this is high or low?
|
- It is not high or low. It can be interpreted as the similarity between groups. - Greater distance level means the groups are more dissimilar. Group 4 is more different from group 2 and 3 because group 4 was separated at distance level 20 while groups 2 and 3 were separated at distance level 13. |
L179-180. Where are these groups? A1, A2, A3, A4... I am trying to find but I do not see in any of your figures.
|
Sorry for confusing. We modified Figure 2. A represents groups in “A”ccretion sites. |
Figure 2. Lacking info. (a) and (b) in the caption. Also, the axis 0 to 25 is the distance level? If yes, please include it in the text or in the figure.
Where are the groups A1, A2, A3, A4 and etc???? Maybe you want to say D1, D2... instead of A1, A2.. ? Also, Where are these groups in the geographical space? Site 1, Site 2...
|
Thank you very much for catching this and sorry for confusing. We modified the Figure 2. - Title for Axis (distance level) was added. - A represents groups in “A”ccretion sites. D1, D2, .. were replaced as A1, A2, …, respectively. - Site 1, Site 2, those were explained in Figure title and S, I and L were explained in figure legends. |
L195 I do not see any sense in this sentence
|
Thank you for catching this. It was changed to canopy layer throughout the MS. |
L197-198 I still do not know where are these groups, A1, A2...
|
Groups were shown in Figure 2 and were explained in L 203-209. |
L 198-199 What do you mean by importance value? Are you sure that it is the right term?
|
- Importance Value is a measure of how dominant a species is in a given community and used to describe species composition in ecology and forestry. - Importance value is explained in methods. L 184-186. |
L242 There are species with very small DBH (10-15cm). In this sense, I have a question. Can this species appear after the accretion process? Maybe the accretion is not influencing the species distribution.
|
We are not sure if trees with DBH < 15 cm were established before or after the accretion. It probably depends on the site. However, our result indicated that small trees in (a) group A1 and (b) A2 in Figure 4 might be established during/after accretion because group 1 and 2 consisted of seaward plots or plots in site 1 (most recent accreted plots). |
L231 I am still lost. I do not know where these groups are.
|
Groups were the result of hierarchical cluster analysis. The plots belonging to each group were shown in Figure 2. |
L231 distribution -> individuals were under...
|
changed. L259. |
L289 stand -> Mangrove...
|
changed. L313. |
L299 this is not everywhere. In Brazil, the Rhiphorora can be often found in higher intertidal areas than Avicennia.
|
The sentence was changed to compare A. alba and R. apiculata. L322. |
L299 rapidly? how fast? years? I think this is not a good word here.
|
“rapidly” was deleted. L331. |
L 300 how about the other species?
|
R. apiculata has tendency to reproduce through propagules. We want to stress Avicennia. L331. |
L300 Rhizophora mangle in the Atlantic-East-Pacific (AEP) are usually at intertidal zones in more unstable environments than Avicennia shauerianna. So, when you make this affirmation it is not true for all species in these genus.
for instance: https://link.springer.com/article/10.2307/1351590
http://www.seer.ufu.br/index.php/caminhosdegeografia/article/view/30856
|
We changed it to “R. apiculata”to make our results clear. L332. |
L 301 all genus of Avicennia or only marina?
|
We changed it to A. marina and A. officinalis. L329. |
L325 I could not give comments here because I still do not know where the A1 and A2... and etc.
|
- Sorry for confusing with errors in Figures and thank you very much for pointing this out. - Groups A1, A2, .. were explained in Figure 2 and relevant contents. |
L388 do you have future recommendations for restoration? What about the other species that you also study?
|
- We added the suggestions for restoration L 432-436 - We did not mention particular species except A. alba and R. apiculata, however, stressed out species composition and the importance of species selection in conclusions. |
L389 if you started with accretion, you should finish with the same word.
|
Thank you very much for checking this. Modified throughout the MS. |

Round 2
Reviewer 2 Report
The revision is much improved. My only concern is in Table 1, based on what definition or criteria, the author define sites that are 'high accesion', 'low erosion' and 'very low erosion'. This is very important because your methods should allow other to follow exactly how to define them. You need to explain based on what criteria to define these groups of mangroves. I think this must be addressed clearly before acceptance of publication.
Author Response
May 27, 2020
Responses to Reviewer Comments
Manuscript ID: Forests-802322 R1
Response to Reviewer 2
The revision is much improved. My only concern is in Table 1, based on what definition or criteria, the author define sites that are 'high accretion', 'low erosion' and 'very low erosion'. This is very important because your methods should allow other to follow exactly how to define them. You need to explain based on what criteria to define these groups of mangroves. I think this must be addressed clearly before acceptance of publication.
|
Thank you very much for catching this. It was relative term among our study sites. We deleted high/low/.. to reduce confusion. The readers can still see the degree of accumulation/erosion by rate in table 1. |
Reviewer 3 Report
Dear Authors,
I am glad about the responses and the changes in your manuscript.
I've just few comments. First, pay attention to the format and final edition, because figures and legend are not in the same page, and lines 97 and 193 have the title section but with the text on different pages.
Best regards
Author Response
I've just few comments. First, pay attention to the format and final edition, because figures and legend are not in the same page, and lines 97 and 193 have the title section but with the text on different pages.
|
Thank you for the comments. We modified the breakage betwween pages. We will be careful that a whole table or figure set will be on the same page at the final editing. |